# Dynamic Analysis and Experiment of a Space Mirror Based on a Linear State Space Expression

**Ruijing Liu** [1,2,3], **Zongxuan Li** [1,3,*], **Wei Xu** [1,3,*], **Xiubin Yang** [1,3], **Defu Zhang** [1,3], **Zhao Yao** [4] **and Kun Yang** [4]

1   Changchun Institute of Optics, Fine Mechanics and Physics, Chinese Academy of Sciences, Changchun 130033, China; liuruijing17@mails.ucas.ac.cn (R.L.); yangxiubin@ciomp.ac.cn (X.Y.); zhangdf@ciomp.ac.cn (D.Z.)
2   University of Chinese Academy of Sciences, Beijing 100049, China
3   Key Laboratory of Space-Based Dynamic & Rapid Optical Imaging Technology, Chinese Academy of Sciences, Changchun 130033, China
4   Army Academy of Armored Forces, Beijing 100072, China; yaozhao8883@gmail.com (Z.Y.); yangkun3338@gmail.com (K.Y.)
*   Correspondence: lizx@mail.ustc.edu.cn (Z.L.); xwciomp@126.com (W.X.)

**Abstract:** Dynamic analysis of the optical–mechanical structure based on the linear state space expression is performed for a three-point flexural mount lightweight space mirror with a diameter of $\phi$740 mm. Using linear structure dynamics and linear state space theories, the state space model of the mirror assembly is established based on modal information. The DC gain method is used to reduce modes and a frequency response analysis of the reduced modes is performed to obtain the frequency domain transfer function between the excitation input and response output points and determine the contribution of each mode to the total frequency response. The frequency response curve is plotted. A mechanical vibration test is performed to verify the accuracy and rationality of simulation analysis. The dynamic analysis method based on state space theory provides a new method of investigating optical and mechanical structures, which can help efficiently and accurately analyze the frequency response characteristics of complex linear systems.

**Keywords:** state space; space mirror; finite element analysis; structural dynamics; vibration test

## 1. Introduction

With the increasingly important applications of modern space optical remote sensing imaging technology in civilian, military and commercial fields, there is a demand for optical systems with better resolution. Increasing the focal length of the optical system is the most direct way to improve its resolution, but this increases the system's aperture. A lightweight and large-aperture primary mirror is the most important optical component of a high-resolution space-to-ground remote sensing optical system. It needs to withstand complex and severe static during processing, inspection, assembly, transportation, dynamic loading, launch, and on-orbit work [1]. Under the above dynamic excitation, the large-scale structural dynamic response of the mirror causes changes in the internal stress state of its assembly structure, such as micro displacements, micro yield and internal friction, resulting in tilt and eccentricity of the mirror surface, deterioration of mirror surface accuracy and other issues. As a result, wave-front difference degradation and visual axis jitter error of the optical system occur, which affect its imaging quality and positioning accuracy; the entire optical system may even completely fail to image [2]. Therefore, it is necessary to conduct structural dynamic analysis of the entire main mirror assembly, evaluate its dynamic response characteristics and ensure that it has good dynamic characteristics.

Modal and frequency response analyses of mirrors are important ways to study the dynamic characteristics of their structures. Modal analysis can help directly observe the vibration form of the mirror, which is the basis for structural dynamic analysis. The

frequency response characteristics of the mirror can directly reflect the response of the mirror structure to external excitation, which is the focus of structural dynamic analysis. Accurate structural dynamic analysis results are based on accurate system modeling. The commonly used modeling methods for spacecraft dynamics modeling include distributed parameter, discrete coordinate, mixed coordinate, modal synthesis, and finite element methods [3]. Among these, the finite element method is most commonly used for structural dynamic analysis of space optical remote sensors. The finite element method is a general numerical method based on the variational method of solving boundary value problems of differential equations and on numerical approximation and discretization to solve the solution structure. Because of its good versatility and self-adaptability, the finite element method is becoming increasingly important for analyzing the dynamic characteristics of an optical system in scientific research and engineering analysis and also promotes the development of finite element analysis software.

To improve computational efficiency, a simplified finite element model is usually proposed, as shown in Figure 1. The optical element is represented by a single node, but the single-point model can only predict the rigid body displacement of the optical element with three translational degrees of freedom and cannot represent the three rotational freedoms and its own elastic deformation. The calculation of shell element modeling is relatively short, but an important limitation of 2D element optical models is that they cannot predict the deformation effect through the optical thickness direction. Accurate system modeling is the premise to ensure the accuracy of analysis results. For lightweight mirror components with complex structures, 3D entity unit modeling is more realistic and accurate to analyze the results, but the scale of analysis and calculation is huge.

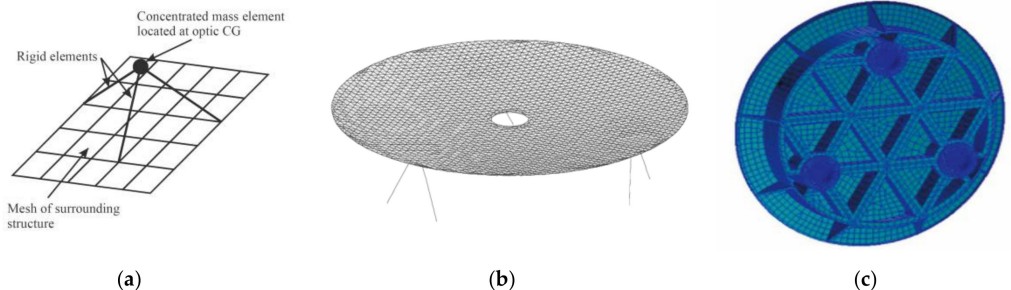

| (a) | (b) | (c) |

**Figure 1.** Simplified method of finite element modeling of a space mirror: (**a**) single-point model, (**b**) 2D shell model, (**c**) 3D solid model.

Most scholars still use finite element analysis to analyze the dynamic characteristics of various structures [4–7]. With the development of finite element technology, various finite element software packages have emerged [8,9]. For systems with complex structures and huge model sizes, such as space optical mirror components, finite element method analysis has the problems of a large calculation volume and low solution efficiency. The simplified model method cannot accurately reflect the true response of the system to external excitations and cannot really meet the needs of engineering practice. To improve the solution rate and simplify the solution method, the state space method can be used to solve the dynamic characteristics of the system. Wu et al. proposed a new method of calculating the dynamic response of a linear system using multiple exponential damping models by introducing higher-order derivatives of the displacement vector as the state variable [10]. Based on the extended state space formalism, an implicit time integration method using linear approximation was developed to analyze the dynamic response of multiple non-viscous damping models of structural systems [11]. Ma used the state space method to solve the eigenvalues and response of the fluid-structure-coupled vibration system [12]. Lezgy-Nazargah et al. used an effective simplified modal state space method to predict the dynamic response of sandwich beams at a low computational cost [13].

Zhe et al. derived a new extended state space method for damping systems and calculated the transient dynamic responses for structural systems using multiple damping models [14].

In recent years, state space theory has been increasingly applied to the fields of mechanics of materials and structural mechanics. The state space method can describe the relationship between input and output, revealing internal system features. In addition, the using vector-matrix and other powerful mathematical tools on the computer to solve more complex problems reduces the dependence on the designer's experience. Therefore, the analysis method based on state space theory has become an important tool for researching the dynamic system [12]. Based on modal analysis and state space theory, this paper provides a method of analyzing the frequency transfer characteristics of various complex structures. Dynamic analysis is performed on a spatial optical mirror with an diameter of ϕ740 mm, modal analysis on the primary mirror assembly is performed using the finite element method, modal information such as the natural frequency of the mirror assembly is obtained, and the single input and single output of the mirror assembly state space model are established. The DC gain method is used to reduce the modes to simplify the model. The frequency transfer characteristics from the input to the output point are analyzed, and the system's frequency response characteristic curve is drawn. Finally, a mechanical vibration test is performed to verify the accuracy of the simulation analysis.

## 2. Theoretical Background

The essence of modal analysis is the linear transformation of coordinates. The coupled equations of motion of the vibration system described by physical coordinates and physical parameters are decoupled by coordinate transformation and transformed into a set of mutually independent equations. Thus, the single-degree-of-freedom solution method can be used to solve the multi-degree-of-freedom system. The decoupled equations can be further reduced and expressed in the form of state space [15].

### 2.1. Decoupling of Equations of Motion

The classical differential equation of the system dynamics model with $n$ nodes, each with six degrees of freedom is as follows:

$$M\ddot{z} + C\dot{z} + Kz = F(t) \tag{1}$$

where $M$, $C$, and $K$ are the mass, damping, and stiffness matrices of the structure, respectively, and their dimensions are all $6\,n \times 6\,n$. $F(t)$ is an input vector with dimensions of $6\,n \times 1$ and $\ddot{z}/\dot{z}/z$ are acceleration, velocity, and displacement vectors, respectively, with the dimensions of $6\,n \times 1$.

When the system is in undamped free vibration, its dynamic model is rewritten as follows:

$$M\ddot{z} + Kz = 0 \tag{2}$$

Assuming a small deformation of the structure, the structure vibrates slightly near the equilibrium position and the solution of the above equations can be set as follows:

$$z_i = z_{mi}sin(w_i t + \Phi_i) \tag{3}$$

where $z_i$ is the displacement vector of all degrees of freedom at the $i$th order frequency, $z_{mi}$ is the $i$th order eigenvector, $w_i$ is the $i$th order eigenvalue, and $\phi_i$ is the initial phase.

Substituting Equation (3) into Equation (2):

$$M\left[-w_i^2 z_{mi}sin(w_i t + \phi_i)\right] + K[z_{mi}sin(w_i t + \phi_i)] = 0 \tag{4}$$

Through the above solution, the eigenvalue $w_i$ ($i$ = 1, ..., m) can be obtained. The corresponding characteristic column vector $z_{mi}$ = $(z_{m1} \ldots z_{mm})$ is the inherent matrix of the system.

Taking a three-degree-of-freedom system as an example, the equation for the *i*th-order eigenvalue is as follows [16]:

$$
\begin{bmatrix}
z_{1i} \\
\dot{z}_{1i} \\
z_{2i} \\
\dot{z}_{2i} \\
z_{3i} \\
\dot{z}_{3i}
\end{bmatrix}
= z_{mi} \sin(w_i t + \phi_i)
\tag{5}
$$

Generalized coordinates $z_1(t)$, $z_2(t)$, and $z_3(t)$ are used to represent the horizontal displacements of three discrete masses $m_1$, $m_2$, and $m_3$, respectively. $F_1$, $F_2$, and $F_3$, respectively, represent the corresponding generalized external forces (one unit acceleration is applied to one degree of freedom at a time, while the other degrees of freedom remain in state). The three-degree-of-freedom system is shown in Figure 2. Three discrete mass motion equations are written using Newton's law.

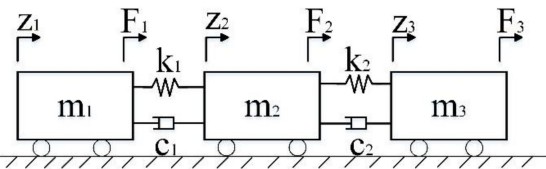

**Figure 2.** A three-degree-of-freedom system.

The mass, damping, and stiffness matrices of the system are obtained by Equations (7) and (8). The equation of motion can be written in the form of Equation (6):

$$
\begin{bmatrix}
m_1 & 0 & 0 \\
0 & m_2 & 0 \\
0 & 0 & m_3
\end{bmatrix}
\begin{bmatrix}
\ddot{z}_1 \\
\ddot{z}_2 \\
\ddot{z}_3
\end{bmatrix}
+
\begin{bmatrix}
c_1 & -c_1 & 0 \\
-c_1 & (c_1 + c_2) & -c_2 \\
0 & -c_2 & c_2
\end{bmatrix}
\begin{bmatrix}
\dot{z}_1 \\
\dot{z}_2 \\
\dot{z}_3
\end{bmatrix}
+
\begin{bmatrix}
k_1 & -k_1 & 0 \\
-k_1 & (k_1 + k_2) & -k_2 \\
0 & -k_2 & k_2
\end{bmatrix}
\begin{bmatrix}
z_1 \\
z_2 \\
z_3
\end{bmatrix}
=
\begin{bmatrix}
F_1 \\
F_2 \\
F_3
\end{bmatrix}
\tag{6}
$$

To simplify the system, let $m_1 = m_2 = m_3 = m$, $k_1 = k_2 = k$, and $c_1 = c_2 = 0$. According to modal analysis theory, we can calculate the eigenvalues $w_1^2 = 0$, $w_2^2 = \frac{k}{m}$, and $w_3^2 = \frac{3k}{m}$. After the eigenvalues are calculated, eigenvectors $z_1$, $z_2$, and $z_3$ are obtained according to Equation (4):

$$
\frac{z_2}{z_1} = \frac{k - w_i^2}{k}
\tag{7}
$$

$$
\frac{z_3}{z_1} = \frac{m^2 w_i^2 - 3kmw_i^2 + k^2}{k^2}
\tag{8}
$$

Each column is the modal vector $z_1$, $z_2$, and $z_3$. To realize the decoupling of the power equation and obtain the contribution value of each mode to the response, both Ansys and MSC Patran adopt the diagonalization and normalization of the mass matrix. Accordingly, the modal, mass, and stiffness matrices are shown in Equations (9)–(11):

$$
z_n = \frac{1}{\sqrt{m}}
\begin{bmatrix}
\frac{1}{\sqrt{3}} & \frac{1}{\sqrt{2}} & \frac{1}{\sqrt{6}} \\
\frac{1}{\sqrt{3}} & 0 & \frac{-2}{\sqrt{6}} \\
\frac{1}{\sqrt{3}} & \frac{-1}{\sqrt{2}} & \frac{1}{\sqrt{6}}
\end{bmatrix}
\tag{9}
$$

$$
m_n = z_n^T m z_n =
\begin{bmatrix}
1 & 0 & 0 \\
0 & 1 & 0 \\
0 & 0 & 1
\end{bmatrix}
\tag{10}
$$

$$k_n = z_n^T k z_n = \begin{bmatrix} 0 & 0 & 0 \\ 0 & \frac{k}{m} & 0 \\ 0 & 0 & \frac{3k}{m} \end{bmatrix} \tag{11}$$

### 2.2. State Space Method Theory

To describe a linear time-invariant system, the corresponding ordinary differential equation or transfer function model of the system under study should be established. However, ordinary differential equations or transfer function models can only describe the relationship between the input and output of the system but cannot describe its internal variables, that is, they cannot contain all the information about the system under study. To make up for the deficiency of ordinary differential equations or transfer function models and show the whole motion state of the system, a finite element modeling method is proposed. However, this method depends on empirical knowledge, and there is always some error with actual test results. To describe the studied system accurately, a state space model is proposed. Using the state space description and the matrix vector method to perform efficient mathematical calculation can be convenient in MATLAB/Simulink and other simulation environments [15]. The linear system state space expression is as follows:

$$\dot{x} = Ax + Bu$$
$$y = Cx + Du \tag{12}$$

Here, $x$ is the state vector, $\dot{x}$ is the derivative of the state vector, $A$ is the system matrix, $B$ is the input matrix, $C$ is the output matrix, $D$ is the feedback matrix, $u$ is the input of the system, and $y$ is the output of the system.

According to the principle of drawing the structure diagram of a univariate system, a general linear system can be represented by the form in Figure 3.

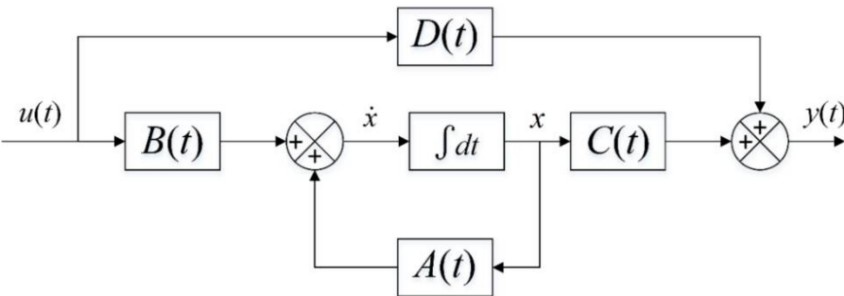

**Figure 3.** State space system block diagram.

The state vector x can be determined as follows:

$$x = \begin{bmatrix} \dot{z} \\ \ddot{z} \end{bmatrix} \tag{13}$$

After the transformation to the principal coordinate, it can be expanded:

$$\ddot{z}_{p1} = F_{P1}$$
$$\ddot{z}_{p2} + + \frac{k}{m} z_{p2} = F_{P2}$$
$$\ddot{z}_{p3} + \frac{3k}{m} z_{p3} = F_{P3} \tag{14}$$

where $F_{pi} = z_n^T F_i$.

Transformation to the principal coordinate system solves the Laplace transform:

$$\boldsymbol{z_p} = \begin{bmatrix} z_{p1} \\ z_{p2} \\ z_{p3} \end{bmatrix} = \begin{bmatrix} \frac{F_{p1}}{s^2+w_1^2} \\ \frac{F_{p2}}{s^2+w_2^2} \\ \frac{F_{p3}}{s^2+w_3^2} \end{bmatrix} = \begin{bmatrix} \frac{z_{n11}F_1+z_{n21}F_2+z_{n31}F_3}{s^2+w_1^2} \\ \frac{z_{n12}F_1+z_{n22}F_2+z_{n32}F_3}{s^2+w_2^2} \\ \frac{z_{n13}F_1+z_{n23}F_2+z_{n33}F_3}{s^2+w_3^2} \end{bmatrix} \tag{15}$$

where $z_{nij}$ is the $i$th row and $j$th column element of the modal matrix. By replacing the inverse principal coordinate system with the physical coordinate system, the contribution values of each mode can be obtained as follows:

$$\boldsymbol{z_p} = \boldsymbol{z_n}^{-1}\boldsymbol{z} \tag{16}$$

For a single-input, single-output damped system:

$$\frac{z_j}{F_k} = \sum_{i=1}^{m} \frac{z_{nij}z_{nki}}{s^2 + 2\xi_i w_i s + w_i^2} \tag{17}$$

where $k$ is the free applied degree of freedom and $j$ is the displacement output degree of freedom and the system damping value.

For a system with critical damping, the motion equation in the principal coordinate system can be expanded as follows:

$$\begin{aligned} \ddot{z}_{p1} &= F_{p1} \\ \ddot{z}_{p2} &= F_{p2} - w_2^2 z_{p2} - 2\xi_2 w_2 \dot{z}_{p2} \\ \ddot{z}_{p3} &= F_{p3} - w_3^2 z_{p3} - 2\xi_2 w_2 \dot{z}_{p3} \end{aligned} \tag{18}$$

where $w_1$, $w_2$, and $w_3$ are the three eigenvalues and $\xi_1$, $\xi_2$, and $\xi_3$ represent the percentages of critical damping for each of the three modes, all of which can be different and are typically obtained from experimental results.

The equation of state is as follows:

$$\begin{aligned} x_1 &= z_{p1} \\ x_2 &= \dot{z}_{p1} \\ x_3 &= z_{p2} \\ x_4 &= \dot{z}_{p2} \\ x_5 &= z_{p3} \\ x_6 &= \dot{z}_{p3} \end{aligned} \tag{19}$$

$$\begin{bmatrix} \dot{x}_1 \\ \dot{x}_2 \\ \dot{x}_3 \\ \dot{x}_4 \\ \dot{x}_5 \\ \dot{x}_6 \end{bmatrix} = \begin{bmatrix} 0 & 1 & 0 & 0 & 0 & 0 \\ 0 & 0 & 0 & 0 & 0 & 0 \\ 0 & 0 & 1 & 0 & 0 & 0 \\ 0 & 0 & -w_2^2 & -2\xi_2 w_2 & 0 & 0 \\ 0 & 0 & 0 & 0 & 0 & 1 \\ 0 & 0 & 0 & 0 & -w_3^2 & -2\xi_3 w_3 \end{bmatrix} \begin{bmatrix} x_1 \\ x_2 \\ x_3 \\ x_4 \\ x_5 \\ x_6 \end{bmatrix} + \begin{bmatrix} 0 \\ F_{p1} \\ 0 \\ F_{p2} \\ 0 \\ F_{p3} \end{bmatrix} u \tag{20}$$

In the output equation under principal coordinates, the number of columns of the output matrix $C$ is the same as the number of state variables.

$$\boldsymbol{y_p} = \begin{bmatrix} 1 & 0 & 0 & 0 & 0 & 0 \\ 0 & 1 & 0 & 0 & 0 & 0 \\ 0 & 0 & 1 & 0 & 0 & 0 \\ 0 & 0 & 0 & 1 & 0 & 0 \\ 0 & 0 & 0 & 0 & 1 & 0 \\ 0 & 0 & 0 & 0 & 0 & 1 \end{bmatrix} \begin{bmatrix} x_1 \\ x_2 \\ x_3 \\ x_4 \\ x_5 \\ x_6 \end{bmatrix} + \begin{bmatrix} 0 \\ 0 \\ 0 \\ 0 \\ 0 \\ 0 \end{bmatrix} u \tag{21}$$

The output equation under the main coordinate is inverse-transformed back to the physical coordinates:

$$\begin{bmatrix} z_1 \\ \dot{z}_1 \\ z_2 \\ \dot{z}_2 \\ z_3 \\ \dot{z}_3 \end{bmatrix} = z_n y_p = \begin{bmatrix} z_{n11} y_{p1} + z_{n12} y_{p3} + z_{n13} y_{p5} \\ z_{n11} y_{p2} + z_{n12} y_{p4} + z_{n13} y_{p6} \\ z_{n21} y_{p1} + z_{n22} y_{p3} + z_{n23} y_{p5} \\ z_{n21} y_{p2} + z_{n22} y_{p4} + z_{n23} y_{p6} \\ z_{n31} y_{p1} + z_{n32} y_{p3} + z_{n33} y_{p5} \\ z_{n31} y_{p2} + z_{n32} y_{p4} + z_{n33} y_{p6} \end{bmatrix} \tag{22}$$

At this point, the output matrix in physical coordinates is:

$$y = Cx = \begin{bmatrix} z_{n11} & 0 & z_{n12} & 0 & z_{n13} & 0 \\ 0 & z_{n11} & 0 & z_{n12} & 0 & z_{n13} \\ z_{n21} & 0 & z_{n22} & 0 & z_{n23} & 0 \\ 0 & z_{n21} & 0 & z_{n22} & 0 & z_{n23} \\ z_{n31} & 0 & z_{n32} & 0 & z_{n33} & 0 \\ 0 & z_{n31} & 0 & z_{n32} & 0 & z_{n33} \end{bmatrix} \tag{23}$$

Since there is no feedback from the system, the feedback matrix $D = 0$. The above theory demonstrates that the coupled vibration equations of a multi-degree-of-freedom system are decoupled by modal coordinate transformation and converted to state space form. The obtained single-degree-of-freedom solution is used as the coefficient to super-position the various modes of the structure. Finally, the total frequency response of the structure is obtained.

*2.3. The Theory of Modal Reduction*

The purpose of establishing a linear state space model of the mirror structure dynamics is to perform rapid matrix calculation and data interaction. The focus is on accurately characterizing the dynamic response characteristics of the mirror structure. The research problem that must be solved is how to reduce the number of extracted modes and the dynamic model to achieve a limited number of modes that can be accurately and reasonably manipulated. To analyze the frequency response, the system model must be concise enough to accurately describe its dynamic performance. To increase the calculation speed, the model should be reduced as much as possible under the premise of ensuring accuracy [15]. The finite element model of the system has many nodes, and the frequency response analysis only needs to clarify the response relationship between the input and output points of the excitation. Therefore, only the mode shapes of the input and measurement nodes are extracted, and the set of degrees of freedom of the finite element model is reduced to a set containing only the state space excitation input and the degree of freedom of the measurement response.

After modal analysis is performed on the finite element model, the number of modes obtained is the same as the number of degrees of freedom of the system. To further simplify the calculation, some important modes need to be retained for mode reduction. The damped system transfer function is as shown in Equation (17), where m is the total number of modes of the system. Under normal circumstances, each transfer function is a superposition of single-degree-of-freedom systems and the residual of each system is determined by the appropriate input and output eigenvectors. The characteristic frequency is determined by the characteristic value. The complex frequency domain variable $s$ needs to be set to 0 to obtain the response of the $i$th mode in the low-frequency DC domain, that is, the DC gain. The modes are reduced according to the magnitude of the DC gain (DC gain method), that is, the mode with a large DC gain is retained, while the mode with a

small DC gain is discarded to further improve the calculation efficiency. The calculation of DC gain is shown in Equation (24), where $s = jw_i = 0$:

$$\frac{z_{ji}}{F_{ki}} = \frac{z_{nij}z_{nki}}{s^2 + 2\xi_i w_i s + w_i^2} = \frac{z_{nij}z_{nki}}{w_i^2} \tag{24}$$

### 3. Simulation Analysis

The mirror is the core component of a space telescope and the physical carrier of beam convergence, and any disturbance on the mirror is directly brought into the optical system and directly affects the system's imaging quality. Based on modal analysis, state space, and modal reduction theories mentioned in Section 2, modeling and dynamic analysis of the mirror assembly are performed to ensure that the primary mirror has good dynamic characteristics in a harsh dynamic environment such as manufacturing, transport, and launch.

### 3.1. Finite Element Model of Mirror Assembly and Modal Analysis

Modal analysis helps to understand the mirror's structural characteristics by solving the natural frequency of the mirror assembly and its model shape. It is possible to intuitively understand the vibration form of the primary mirror and its supporting structure, which is the basis for analyzing the dynamic characteristics of the structure. This paper uses finite element software programs Hypermesh (Altair, Troy MI, USA) and MSC Patran (MSC Software, Los Angeles CA, USA) to mesh the mirror component structure model, set boundary conditions, and assign element attributes. The mirrored finite element model is shown in Figure 4, with 38,508 elements and 53,797 nodes.

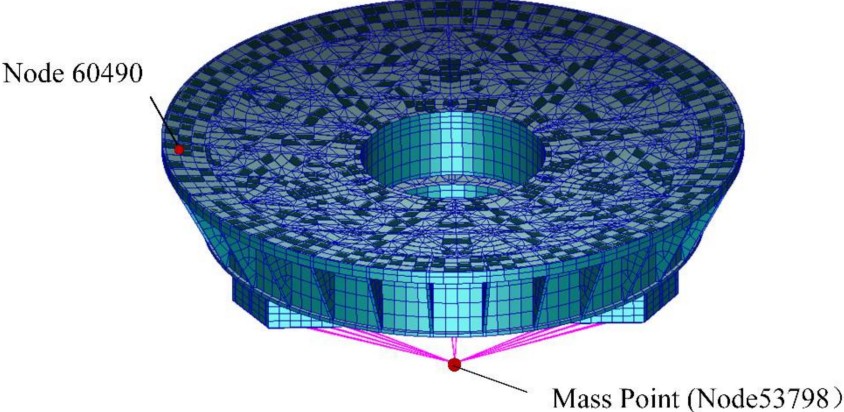

**Figure 4.** Finite element model of the primary mirror assembly.

Modal analysis of the finite element model of the primary mirror is performed using MSC/NASTRAN (MSC Software, Los Angeles CA, USA). To simulate the excitation of the mirror assembly by the shaker in the mechanical vibration test, the large-mass point element method is used for analysis. The large-mass simulation method is used to simulate the shaker in the vibration test. In an on-ground aerospace vibration test, the shaker is excited by an electromagnetic coil and vibrates in only one DOF, constrained by a hydrostatic slide guide. This is simulated in MSC.Patran (MSC Software, Los Angeles CA, USA) by selecting such boundary conditions. To ensure the accuracy of the calculation, the large mass point is $10^3$–$10^8$ times the structural mass. When the large mass point is set to $10^3$ times the mass of the component, applying acceleration on the mass point cannot fully excite the structural response of the mirror body and a rigid body mode occurs. However, when the large mass point is set to $10^4$ times the mass of the component, it can not only stimulate the structural response of the mirror body but also better imitate the excitation of the shaker to the mirror body holder. Therefore, a quality point with a quality $10^4$ times the mass of the mirror body is established. The mass point and the reflector assembly are constrained in the form of

MPC. Considering the rigid connection between the mirror body and the shaker, the RBE2 MPC is used to connect the large mass point unit to the node of the main mirror substrate mechanical installation interface. During the test, the connection relationship between the shaker and the mirror assembly remains unchanged and there is no relative movement between the two. Therefore, the boundary condition is constrained, that is, the large mass point is constrained with 6 degrees of freedom and the first 10 natural frequencies of the mirror are calculated. Figure 5 shows the first six modal shapes of the mirror assembly.

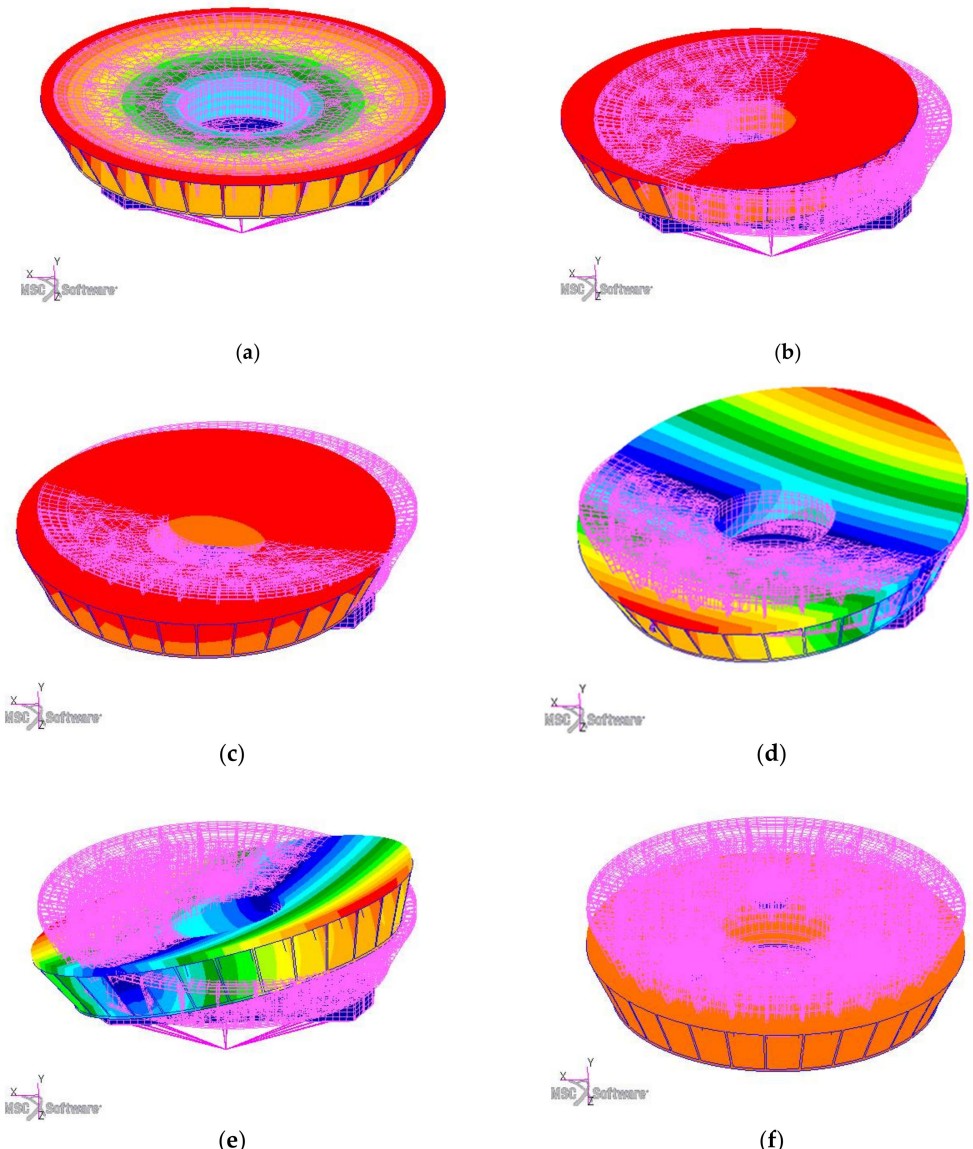

(a)

(b)

(c)

(d)

(e)

(f)

**Figure 5.** Modal shapes of the first 6 modes: (**a**) F1 = 204.35 Hz, rotated about z; (**b**) F2 = 228.95 Hz, trans along x; (**c**) F3 = 228.95 Hz, trans along y; (**d**) F4 = 274.52 Hz, rotated about x; (**e**) F5 = 274.52 Hz, rotated about y; and (**f**) F6 = 318.42 Hz, trans along z.

According to the modal analysis, the stiffness of the main mirror assembly is large and the first mode is 204.35 Hz, which can avoid resonance with other structures of the star in the process of carrier reflection.

### 3.2. Frequency Response Analysis of Finite Element Method

The frequency transfer characteristic of the mirror refers to the function of the maximum steady-state response of the mirror structure to the frequency domain simple har-

monic excitation with a change in the excitation frequency. The frequency transfer characteristic can reflect the response of the mirror structure to the external excitation, which is a structural dynamic analysis. Under normal circumstances, the study of the frequency response characteristics of the mirror relies on the finite element method.

In the finite element method, finite element software is used for frequency response analysis. According to the mirror modal analysis results, the frequency range is set to (220 Hz, 1640 Hz), the frequency step is 20 Hz. Based on the previous engineering experience of the research group and the experimental results, the structural damping was revised. Through the acceleration amplitude-frequency response data of the dynamic experiment, the dynamic amplification factor Q of the system is estimated to be 50.3. Q = 1/ξ (where ξ is the structural damping coefficient), and the structural damping coefficient value g = 0.02 is obtained after calculation. The frequency response analysis of the finite element model of the primary mirror assembly is performed by the modal method in MSC/NASTRAN (MSC Software, Los Angeles CA, USA), that is, the modal mode shape of the structure is used to reduce and decouple the coupled motion equations. Through the established large mass point (node 53,798), the main mirror assembly is excited with a 0.2 *g* acceleration frequency response and the boundary conditions are constrained. The middle position of any two support points of the mirror (i.e., at the cantilever) is selected as the response output point, that is, node 60490. The analysis result is shown in Figure 6.

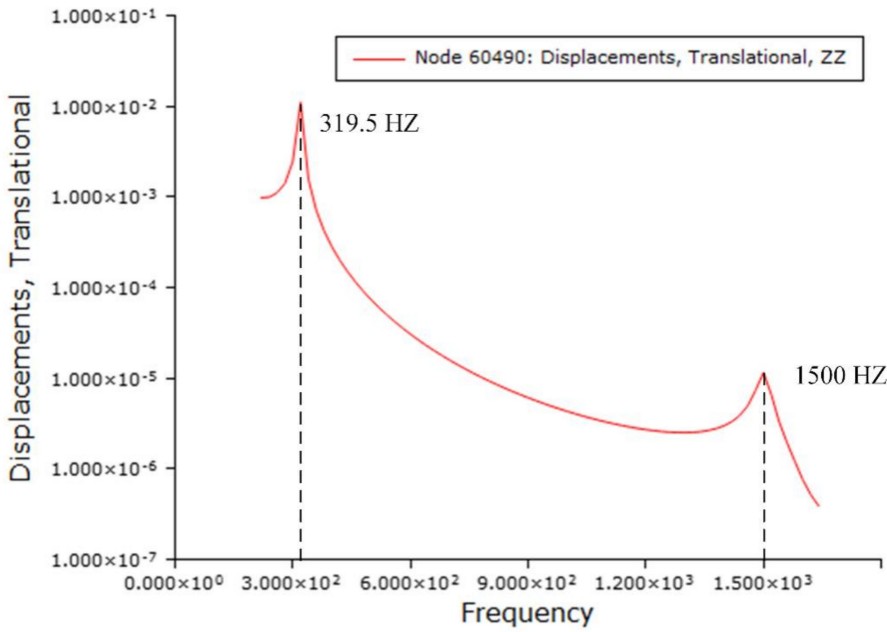

**Figure 6.** Frequency response curve of the primary mirror.

### 3.3. Frequency Response Analysis by the State Space Method

Although finite element software is widely applied to the frequency response analysis, for more complex and large models, the finite element method involves a huge number of calculations and takes time to provide a solution due to the large model dimensions. In addition, the software function is limited, and it is inconvenient to observe the response contribution of each mode, so the finite element method cannot always meet the needs of engineering practice [17]. To efficiently and accurately solve practical engineering problems, the state space method can be used to calculate structural frequency response problems. According to the modal analysis results, the frequency range of frequency response analysis is confirmed to be [220 Hz, 1640 Hz], the damping is in the form of modal damping g = 0.02, and the state space model of the primary mirror assembly is established. Take the frequency transfer characteristics between Node 53,798–Node 60,490 as an example. Node 53,798 is the excitation input point, while node 60,490 is the response output point.

The state space method often uses the scientific computing software MATLAB for analysis and problem solving. The analysis steps are as follows:

(1) Data preparation. Perform modal analysis on the mirror component structure, extract the natural frequency of the appropriate order modal data from the obtained f06 result file; extract the eigenvalues of the system, as shown in Figure 7; extract the force input degrees of freedom, response output degrees of freedom, and form a modal matrix (take the displacement in the Z direction as an example, as shown in Equation (26)); and determine the frequency range based on the results of the modal analysis.

(2) Establish a state space model. Read the data file, use Equations (20)–(23) to establish the state space model of the structure, and draw the total frequency transfer characteristic curve and the frequency transfer characteristic curve of each mode.

(3) Reduce modes. Use Equation (24) to solve the DC gain values of each non-rigid body mode and sort them from largest to smallest. The mode corresponding to the larger DC gain value is retained, while the mode with the smaller DC gain value is discarded, and the state space model is regenerated by the retained mode to obtain a simplified system.

(4) Obtain the result. Draw a Bode diagram of the simplified system to obtain the frequency transfer characteristic curve of nodes 53,798–60,490.

```
                        R E A L   E I G E N V A L U E S
   EIGENVALUE              RADIANS            CYCLES         GENERALIZED        GENERALIZED
                                                               MASS              STIFFNESS

  2.176313E+06          1.475233E+03       2.347906E+02      1.000000E+00       2.176313E+06
  2.468935E+06          1.571285E+03       2.500777E+02      1.000000E+00       2.468935E+06
  2.468942E+06          1.571287E+03       2.500781E+02      1.000000E+00       2.468942E+06
  3.195731E+06          1.787661E+03       2.845151E+02      1.000000E+00       3.195731E+06
  3.195752E+06          1.787667E+03       2.845160E+02      1.000000E+00       3.195752E+06
  4.002834E+06          2.000708E+03       3.184226E+02      1.000000E+00       4.002834E+06
  8.907505E+07          9.437958E+03       1.502098E+03      1.000000E+00       8.907505E+07
  9.875758E+07          9.937685E+03       1.581632E+03      1.000000E+00       9.875758E+07
  9.875802E+07          9.937707E+03       1.581635E+03      1.000000E+00       9.875802E+07
  1.048409E+08          1.023918E+04       1.629617E+03      1.000000E+00       1.048409E+08
  1.048478E+08          1.023952E+04       1.629670E+03      1.000000E+00       1.048478E+08
```

**Figure 7.** F06 data file.

According to mode reduction theory in Section 2.3, the DC gain calculation of each mode is shown in Equation (25) and the order of the modes is based on the DC gain: 6, 1, 7, 3, 2, 9, 5, 10, 4, 8. As can be seen from the analysis results in Figure 8, the DC gain does not increase with the increase in frequency; the DC gain of modes 9, 5, 10, 4, and 8 is small; and the contribution of modes 5, 10, 4, and 8 to the overall frequency response is small. The frequency response analysis curves in Figures 9–11 can also verify this point. It can be seen in Figure 11 that modes 6, 1, 7, 3, and 2, in that order, make a greater contribution, while the mode with a smaller DC gain makes a small contribution to the frequency response function. Figure 12 shows that the frequency response curve of only the 5th-order mode with the largest DC gain is highly consistent with the frequency response curve of all modes. Currently, increasing the number of selected modes does not increase the calculation accuracy and reduces the calculation effectiveness:

$$dc\_gain = |x_n(node\_force, :). \times x_n(node\_dis, :)|./w^2 \tag{25}$$

where $x_n$ is the system modal matrix (as shown in Equation (26), which is composed of the Z direction displacement of the excitation input point and the response output point) and $w = 2\pi \cdot frequency$ can be obtained in the f06 file after modal analysis, as shown in Figure 7.

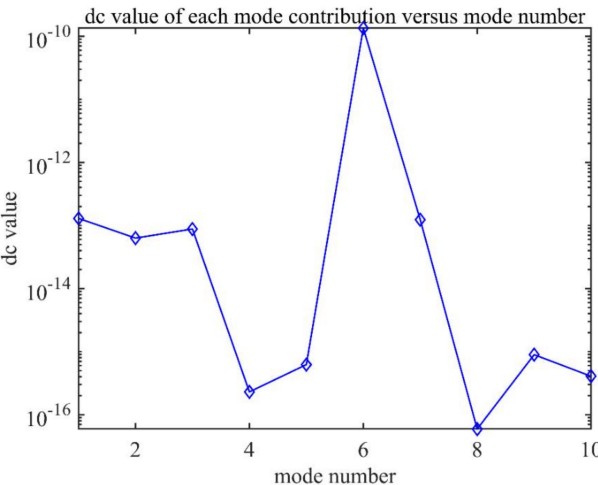

**Figure 8.** DC value of each mode contribution versus mode number.

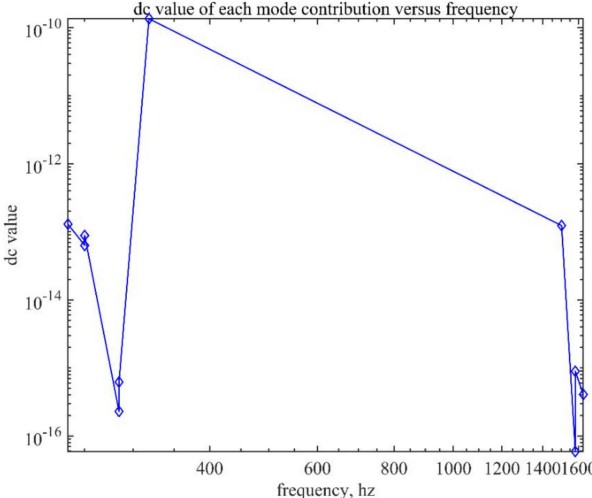

**Figure 9.** DC value of each mode contribution versus frequency.

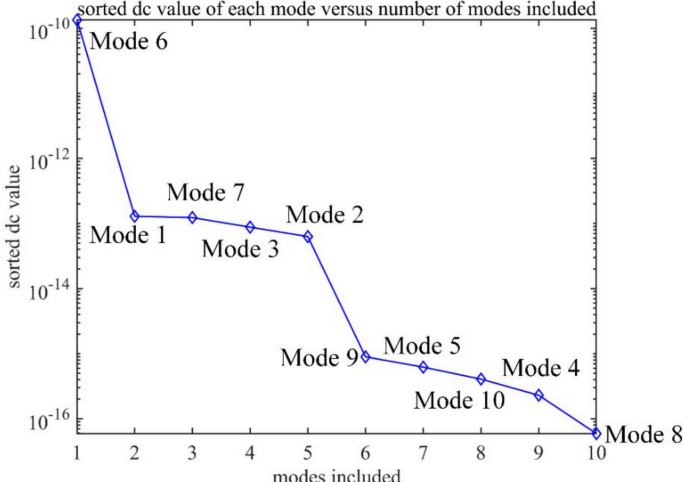

**Figure 10.** Sorted DC value of each mode versus number of modes include.

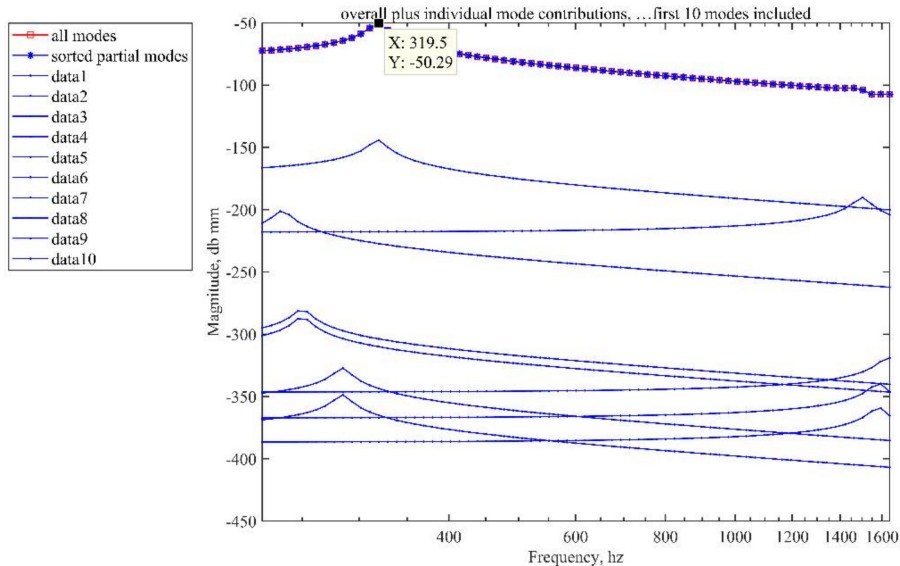

**Figure 11.** Overall plus individual mode contributions for the ten sorted mode model.

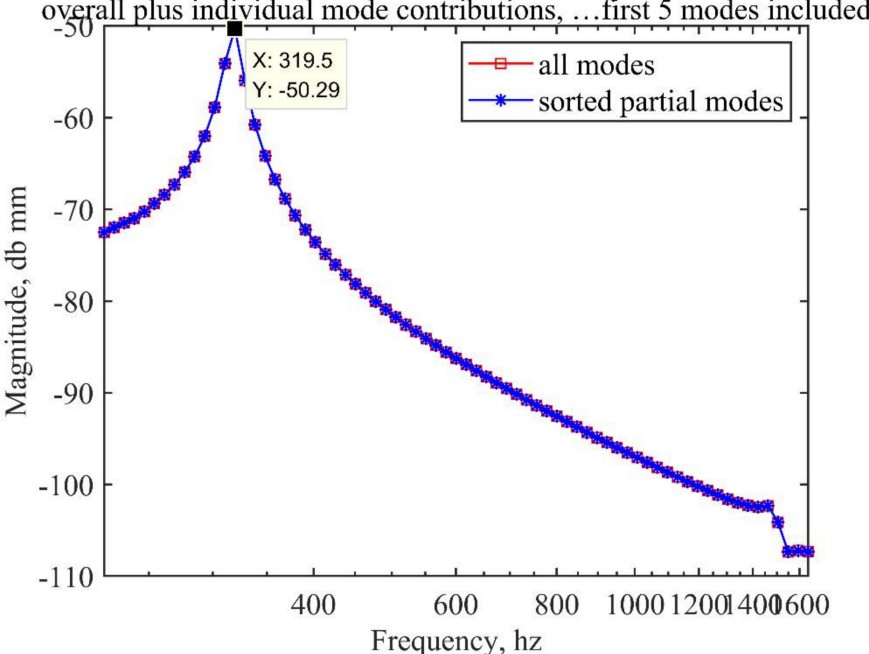

**Figure 12.** Overall plus individual mode contributions, first five modes include.

$$x_n = \begin{bmatrix} Z_{1\_node\_excitory\ input} & \cdots & Z_{n\_node\_excitory\ input} \\ Z_{n\_node\_response\ output} & \cdots & Z_{n\_node\_response\ output} \end{bmatrix} \begin{matrix} \leftarrow DOF_1 \\ \leftarrow DOF_n \end{matrix} \qquad (26)$$

$$\begin{matrix} \uparrow & \uparrow \\ Mode1 & Mode2 \end{matrix}$$

The frequency response curve of node 53,798–60,490 is obtained based on the state space method as shown in the Figure 11.

The state space method and the finite element method were used to analyze the frequency transfer characteristics of the main mirror assembly. The results showed that the linear trend in the frequency response curve within the frequency range is the same and the frequency points corresponding to the peak and trough of the curve are also the same. From the above analysis, the DC gain of mode 6 is the largest, which means that

mode 6 is the main reason for the first peak of the overall frequency response function curve. As shown in Figure 13, the phase curve obtained by the state space method and the frequency response curve obtained by the finite element method do not completely coincide. The state space method performs high-frequency truncation, so only the first 10 modes are truncated for analysis. Starting from the 11th mode, the frequency is much higher than that of the 10th mode and the response of the structural system to high-order frequencies is extremely low, so there is no need for further analysis. The frequency response analysis of the finite element method is performed on the basis of structural modal analysis and contains high-frequency information. Because Patran and the state space method proposed in this paper have different fitting algorithms for the frequency response characteristic curve, the low-frequency response curve has errors. The structure is prone to damage and plastic deformation at the resonance peak, so we only care about the response of the structure at the resonance peak. The relative errors of the frequency response curves obtained by the two methods at the resonance peak response are 11.3% and 0.7% respectively, which are within the allowable range according to engineering experience. Therefore, the analysis of the state-space method is accurate.

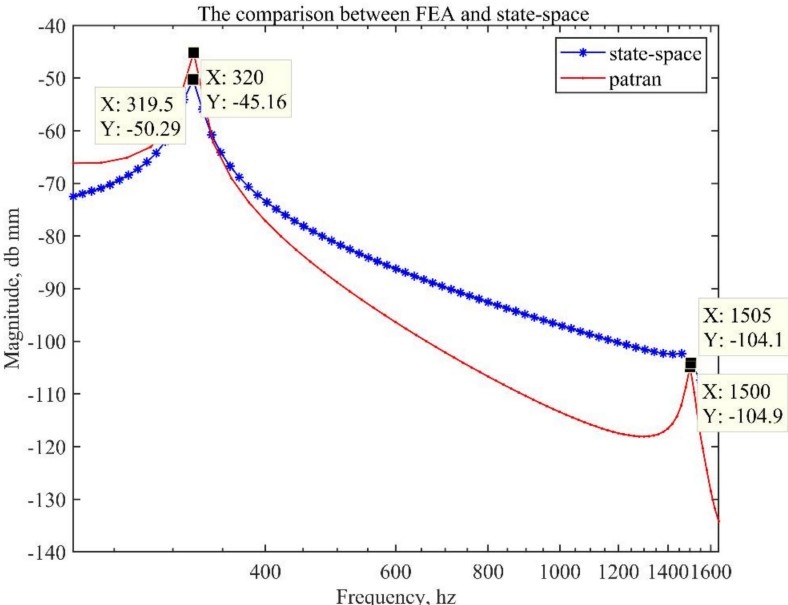

**Figure 13.** Comparison of finite element analysis (MSC.Patran) and the state space method.

The method of analyzing the frequency transfer characteristics of the mirror assembly based on the state space method is accurate. However, it takes about 10 min to obtain finite element analysis results on the frequency response of the primary mirror assembly, while the state space method only needs tens of seconds to provide the frequency response characteristic curve and its calculation efficiency is far more limited. The finite element method is superior, and the contribution value of each mode can be viewed, which is more convenient for engineering analysis.

The finite element method is used to analyze the frequency response characteristics of the structure. There are two kinds of numerical methods to choose: direct frequency response and modal frequency response. The direct frequency response directly solves the coupled motion equations according to the given frequency; while the modal frequency response uses the mode shape of the structure to reduce and decouple the coupled motion equations, a and at the same time, the corresponding results at a given frequency are obtained by the corresponding superposition of a single mode. Since the modal frequency response method can only be used to decouple the equations when there is no damping or only modal damping, this paper uses the direct method to solve the problem. As shown

in Table 1, the state space method simplifies the model by reducing the modal frequency response, which can improve the computational efficiency.

**Table 1.** Comparison of calculation efficiency between FEA and state space.

| Item | Patran | State-Spsce |
|------|--------|-------------|
| Analytical procedures | Create model → Create a frequency dependent load case → Create a frequency dependent field → Define frequency-varying excitation → Submit the file for analysis: Direct frequency response → Obtain the result | Create model → Data preparation: natural frequency, model eigenvalue, modal matrix... → Linear State-Space modal → Model reduction → Obtain the result |
| Elapsed time | 16 min | 20 s |

## 4. Dynamic Vibration Test of Primary Mirror Assembly

To accurately evaluate the dynamic response of the primary mirror assembly in an external dynamic environment, the accuracy of relevant theoretical and simulation analyses can be verified through dynamic tests. The diameter of the primary mirror is ϕ740 mm, its thickness is 150 mm, and its weight is 47 kg (including the primary mirror, the invar insert, the flexural mount, the primary mirror substrate, connecting screws, and the related components of the material and structural form), which is completely consistent with the project design. The primary mirror assembly is connected to the shaker through an aluminum vibrating fixture, whose size is 1000 mm × 1000 mm × 45 mm. The dynamic test is performed at the environmental test station. The mirror assembly and the vibration fixture are installed on an LMS-V964 9-ton electromagnetic shaker (BK Company, Copenhagen, Denmark). During the test, the mirror assembly and the vibration fixture are connected by steel screws. The connection relationship remains unchanged during the transfer operation. The shaker applies excitation to the mirror assembly and then performs sine frequency sweep, sinusoidal vibration, and random vibration tests on the reflector assembly in sequence. The vibration test process is shown in Figure 14.

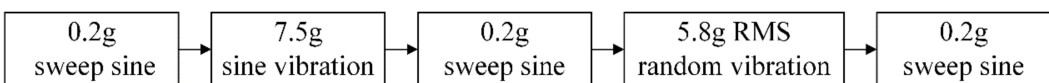

**Figure 14.** Flowchart of the mechanical vibration test.

During the dynamic test, an acceleration sensor is required for measurement, and the sensor bonding position is shown in Figure 15. The 1# sensor is a three-axis acceleration sensor, and its three orthogonal measurement directions are consistent with the coordinate axis directions in Figure 15; 2#–8# are all single-axis acceleration sensors. The 2# sensor is glued to the side of the aluminum triangle plate, while the other sensors are glued to the upper surface of the main mirror. Figure 15 shows the bonding position and the measurement direction of sensors 2#–8# (where the measurement direction of 5# and 8# is parallel to the Z direction, that is, the vertical mirror direction). The acceleration sensors cannot be removed until the vibration test in the X, Y, and Z directions is completed. The test site of the main mirror assembly is shown in Figure 16.

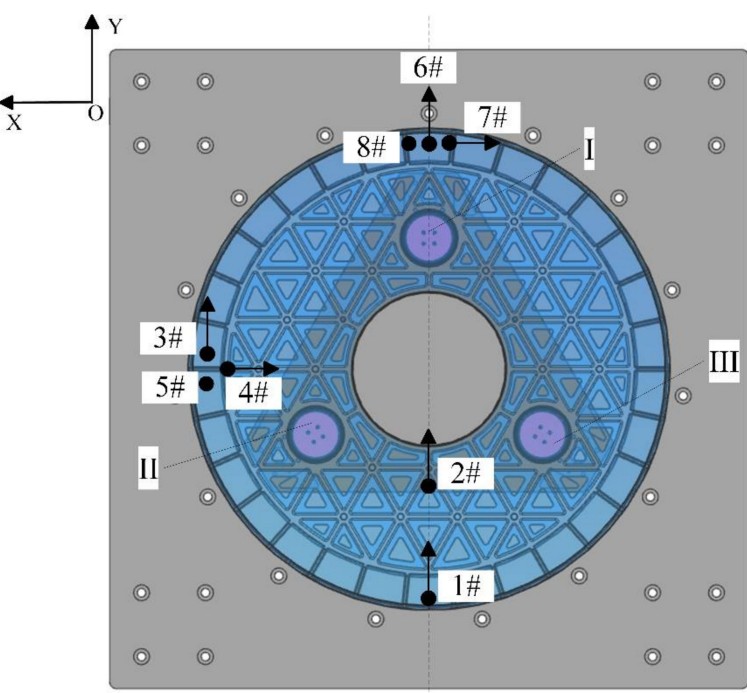

**Figure 15.** Sensor bonding position.

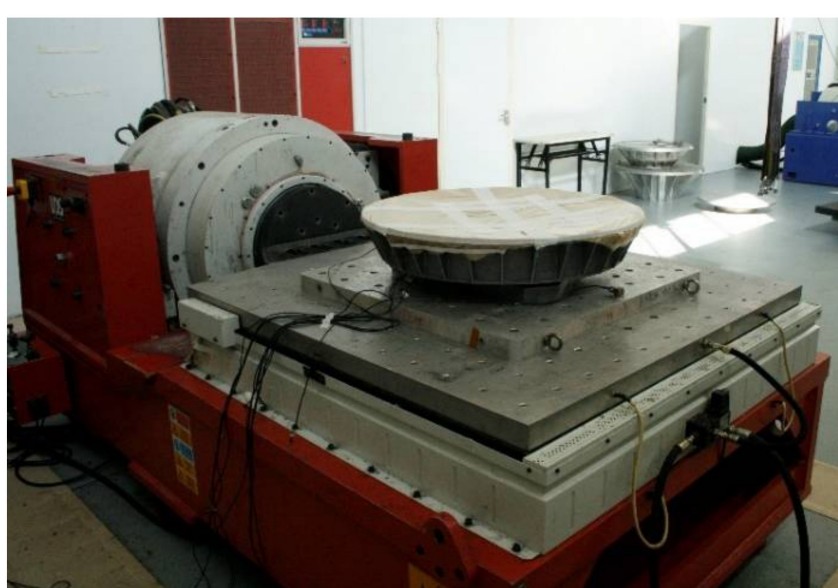

**Figure 16.** Vibration test of the primary mirror assembly.

A frequency sweep test is performed on the reflector in the range of 10–2000 Hz. As shown in Figure 17, errors occur between the theoretical model and the actual structure. However, the frequency response curve and the peak response are the same. The first resonance peak is selected, that is, the moving motion state of the excitation extension in the direction of X/Y/Z, which is the first-order natural frequency. The maximum test value of the first-order natural frequency of the mirror assembly in the X direction is 213.702.

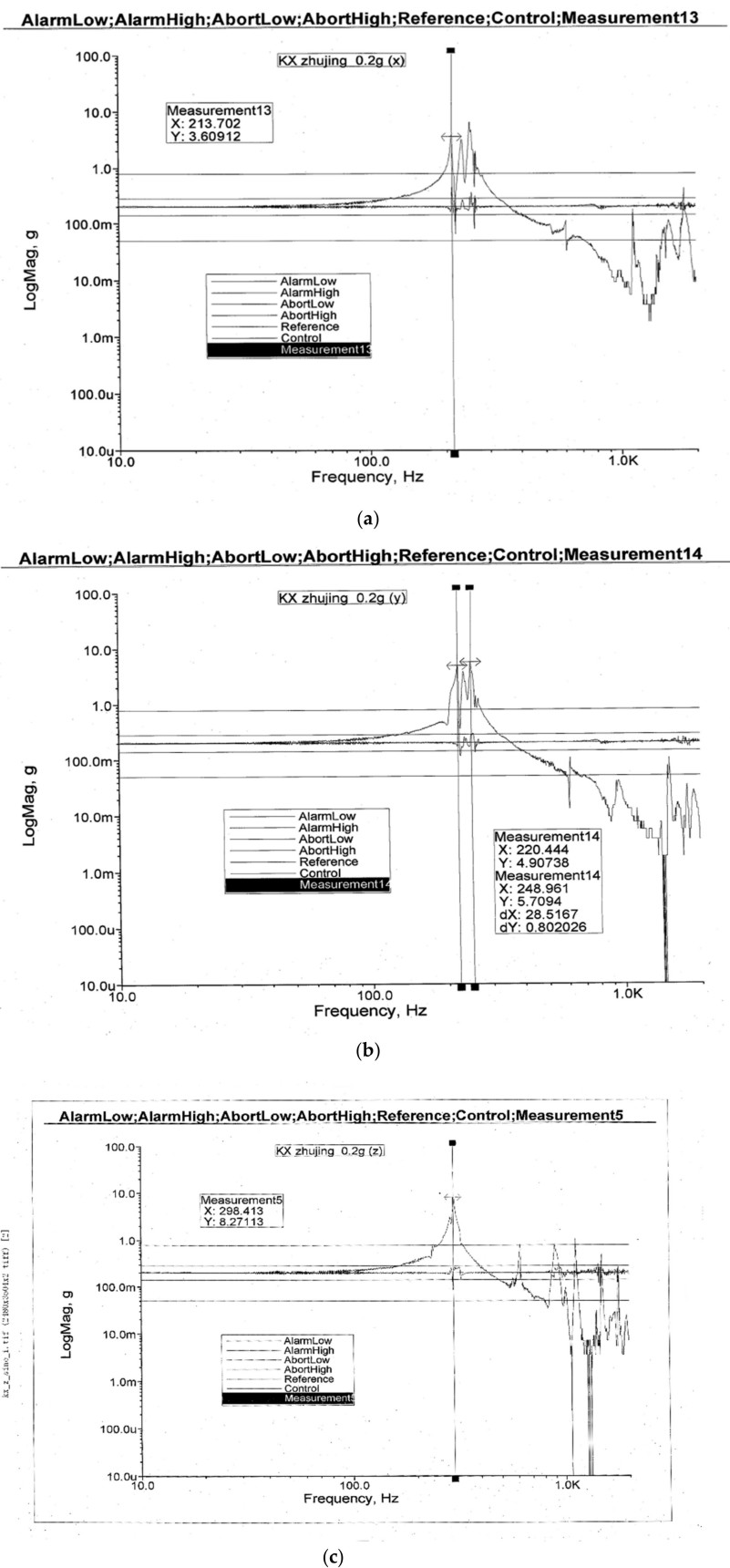

**Figure 17.** Results of the sine sweep test: (**a**) X direction sine sweep test; Fx = 213.702 Hz. (**b**) Y direction sine sweep test; Fy = 220.444 Hz. (**c**) Z direction sine sweep test; Fz = 298.413 Hz.

Hz, the natural frequency of the simulation analysis is 228.95 Hz, and the relative error is 6.7%. The maximum test value of the first-order natural frequency of the mirror assembly in the Y direction is 220.444 Hz, the natural frequency of the simulation analysis is 228.95 Hz, and the relative error is 3.7%. The maximum test value of the first-order natural frequency of the mirror assembly in the Z direction is 298.413 Hz, the natural frequency of the simulation analysis is 318.42 Hz, and the relative error is 6.3%. The trend in the frequency response curve of the simulation analysis is roughly the same as that of the frequency sweep test. The relative error of the natural frequency in the X, Y, and Z directions is small, which proves that the simulation analysis results of the primary mirror assembly are accurate. The connection reliability of the mirror assembly in the three directions is high, and the dynamic stiffness is good.

## 5. Conclusions

Based on modal analysis and state space theories, this paper analyzes the dynamic characteristics of a space mirror with a diameter of $\phi$740 mm. The linear state space model of the mirror assembly is established, the frequency response of the mirror assembly is solved using MATLAB software, and the influence of each mode on the overall response is analyzed. The modes are sorted according to the DC gain, and less important modes are discarded to simplify the model. The results of the reduced-order model and the unsimplified model are close, proving that the method can simplify the calculation and analysis process, while ensuring accuracy. Finally, the structural reliability of the mirror assembly and the accuracy of simulation analysis are verified by the mechanical vibration test of the primary mirror assembly. The simulation results and test results are shown in Table 2, which demonstrates that the accuracy of the simulation analysis in this paper is relatively high. This paper provides a new method of performing dynamic analysis of the opto-mechanical structure. The method is simple and flexible and greatly improves the calculation efficiency, ensuring better accuracy. It can be widely used in engineering practice.

**Table 2.** Comparative analysis of simulation results and test results.

| | X Direction | | Y Direction | | Z Direction | |
|---|---|---|---|---|---|---|
| | Fundamental Frequency (Hz) | Relative Error | Fundamental Frequency (Hz) | Relative Error | Fundamental Frequency (Hz) | Relative Error |
| FEA simulation Test | 218.95 213.702 | 6.7% | 218.95 220.444 | 3.7% | 308.02 298.413 | 6.3% |

**Author Contributions:** R.L. conducted the theoretical derivation and simulation analysis; W.X. and Z.Y. conceived and designed the experiments; D.Z. performed the experiments; R.L. wrote the paper; and Z.L. proofread and revised the paper. Z.Y. and K.Y. conduct research and investigation processes. X.Y. data curation. All authors have read and agreed to the published version of the manuscript.

**Funding:** This research was funded by the National Natural Science Foundation (grant no. 62005275).

**Institutional Review Board Statement:** Not applicable.

**Informed Consent Statement:** Not applicable.

**Data Availability Statement:** Not applicable.

**Conflicts of Interest:** The authors declare no conflict of interest.

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
