# Peer review of "Dynamic Analysis and Experiment of a Space Mirror Based on a Linear State Space Expression"

_applsci, doi:10.3390/app11125379_

Round 1

Reviewer 1 Report

Review report applsci – 1217081

Manuscript title

 Dynamic Analysis and Experiment of Space Mirror Based on Linear State Space Expression

 Authors

Ruijing Liu, Wei Xu, Zongxuan, Xiubin Yang, Defu Zhang

The manuscript examines the dynamic characteristics of a space mirror with a diameter of 740mm using both numerical modelling and simulation and mechanical vibration experimental test. As for theoretical resources, the authors have been applied modal analysis theory and state space theory, which are well known methods, previously used in literature for mechanical system analysis.

Therefore, the present paper could be considered a case study about the structural reliability of a mirror assembly and the accuracy of simulation related to modal analysis and frequency response.

Conceptually, the proposed paper roughly validates a method of dynamic analysis of mechanical systems.

The question is whether the method presented in the paper can be applied in optimization problems, in the design stage.

Firstly, simulation analysis was carried out and finite element software Hypermesh and MSC.Patran was used to mesh the mirror component structure model, set boundary conditions and assign element attributes. Secondly, the modal analysis of the discretized model of the mirror was developed by Msc.Nastran finite element software.

Thirdly, the modal method in MSC/NASTRAN was applied to analyze the frequency response of the FEM assembly.

Generally, the present approach proves a good knowledge of the problem, and describes a methodology dedicated to the dynamic analysis of the space mirror, using efficient algorithm based on finite element analysis.

Even so, in my opinion, the theoretical background depicted in the second paragraph (“Principle of modal analysis”) is too long and should be restrained.

Also, it is not clear for the reader how the frequency response of the mirror assembly was solved by MATLAB software.

Some acronyms should be explained (i.e. RBE2 MPC, DC).

Author Response

Response Letter

Professor Maria Simion

Section Managing Editor

Applied Science

May 14, 2021

Dear Prof. Simion:

Subject: Submission of revised manuscript Dynamic Analysis and Experiment of Space Mirror Based on Linear State Space Expression. (Manuscript ID applcsi-1217081)

Sincerely thank you and the reviewers for the time that you have invested in evaluating our manuscript and the valuable comments. We sincerely apologize for the great time it has taken us to respond to these, and hope that a revised version of the manuscript will still be considered by Applied Science. We have modified the manuscript in response to the extensive and insightful reviewer’s comments. Our responses are given in a point-by-point manner below. Every change that we made to the manuscript are shown in another word file that we uploaded by using yellow-color-highlighted text.

Reviewer #1:

Comments to the Author:

Q1. The question is whether the method presented in the paper can be applied in optimization problems, in the design stage.

Response:

The method proposed in the manuscript can be applied to the optimization problem in the design stage. The method can be used to calculate the frequency response characteristics of the optimized components every time the structure is optimized, which can efficiently and accurately reflect the response of the mirror structure to external excitations.

Q2. Even so, in my opinion, the theoretical background depicted in the second paragraph (“Principle of modal analysis”) is too long and should be restrained.

Response:

According to your comment, I have simplified the theoretical derivation process in the article.

The second part is the theoretical background foundation of the whole manuscript, including modal analysis theory, state space theory, and modal reduction theory. Based on the theory of the second part, the frequency response analysis of the primary mirror assembly is analyzed by MATLAB, so the description is very detailed.

Q3. Also, it is not clear for the reader how the frequency response of the mirror assembly was solved by MATLAB soft-ware.

Response:

According to your comment, the process of analyzing the frequency response of the state space method is described in detail in the manuscript. Based on the theoretical background introduced in the second part, follow the steps of obtaining the frequency response curve introduced in the manuscript and program in MATLAB.

Q4. Some acronyms should be explained.

Response:

It has been modified in the manuscript according to your comments.

Reviewer 2 Report

### REVIEW REPORT ###

MANUSCRIPT ID: applsci-1217081

MANUSCRIPT TITLE: Dynamic Analysis and Experiment of Space Mirror Based on Linear State Space Expression

RECOMMENDATION: Reject

REPORT:
The paper summarizes the results of an experimental/numerical study on structural dynamic analysis of a space mirror. The authors used FE software and linear state space model to estimate Eigen-frequencies of the mirror, its mode shapes and amplitude-frequency characteristics. Results of the simulations are then compared with the results of the dynamic shaker experiment. The paper brings no novelty as its content represents a straightforward engineering approach for dynamic structural analysis of a construction using either commercial FE code or a well-known state space representation. Moreover, the presentation of the results is unclear and quality of the simulations is questionable. Overall quality of the paper, including language style, formatting and clarity for reader, is low. Therefore, I do not recommend the paper for publication.

COMMENTS:
1) English, formatting and spell-check - I went through some typos and formatting issues (e.g., figure captions on different pages than figure itself). More serious problem is the style as it contains a lot of statements that are not clear, have no informative value, are contradicting or not correct (e.g., page 1/lines 43-45, page 3/lines 92-96, page 3/lines 127-129, page 3/lines 133-134, page 6/lines 201-204).
2) Introduction - 1st paragraph - completely missing context. The reader does not know anything about the mirror and the system.
3) Introduction - page 2/lines 57-76 - what is the purpose of this paragraph? There is a lot of references to the papers with the similar topic. The authors should clarify why they include them and how they improve the methods used in the referenced papers.
4) Section 2 - Principle of modal analysis - does not contain anything new. The described methods are well-known while the authors do not clarify what is new in their approach. In fact, the whole section summarizes an usual content of the advanced linear structural dynamic courses at the technical universities.
5) Modal matrix and equations decoupling - the statements about the decoupling are misleading. It is not true that the vibration system can be, in general, decoupled as demonstrated, as the authors do not discuss the problems related to the damping matrix. Even in the example shown in Figure 2, the system can be decoupled only for certain values of damping coefficients that will produce diagonal modal damping matrix. The problem of damping is serious and damping related effects can be observed even in the presented results of the paper and are not discussed.
6) Section 3.1, page 10/lines 320-323 - Boundary conditions and models are not sufficiently described. Why 10e4 point mass (shaker table experiment)? RBE2 - rigid coupling only allowing certain movements of the mirror? Why did the authors selected such boundary conditions?
7) Figure 5 - missing coordinate system.
8) Results - node numbers (FE simulation) are irrelevant for reader as he does not know the model. 
9) Figures 7-13 - obviously, the state space model does not correspond to the FE model - what are the reasons? Numerical damping of the FE simulation? Damping of the state space model? (e.g., high frequency peak is not detectable for state space model - certainly, it is not in the location suggested by the authors).
10) Experimental results - the frequency spectra are very different in comparison to the models. Authors do not discuss anything about it. Quality of the plots is very low. What is the meaning of AlarmLow curve, etc.?
11) Table 1 - what is compared? Which simulation (FE, state space)? How the authors selected the experimental results of Eigen-frequencies? How did they determine the frequency from three coupled peaks in the experimental data?

Author Response

Response Letter

Professor Maria Simion

Section Managing Editor

Applied Science

May 14, 2021

Dear Prof. Simion:

Subject: Submission of revised manuscript Dynamic Analysis and Experiment of Space Mirror Based on Linear State Space Expression. (Manuscript ID applcsi-1217081)

Sincerely thank you and the reviewers for the time that you have invested in evaluating our manuscript and the valuable comments. We sincerely apologize for the great time it has taken us to respond to these, and hope that a revised version of the manuscript will still be considered by Applied Science. We have modified the manuscript in response to the extensive and insightful reviewer’s comments. Our responses are given in a point-by-point manner below. Every change that we made to the manuscript are shown in another word file that we uploaded by using yellow-color-highlighted text.

Reviewer #2:

Comments to the Author:

Q1. English, formatting and spell-check - I went through some typos and formatting issues (e.g., figure captions on different pages than figure itself). More serious problem is the style as it contains a lot of statements that are not clear, have no informative value, are contradicting or not correct (e.g., page 1/lines 43-45, page 3/lines 92-96, page 3/lines 127-129, page 3/lines 133-134, page 6/lines 201-204).

Response:

(1) page 1/lines 43-45: Stated the necessity of frequency response analysis of structural components.

(2) page 3/lines 92-96: Modifications were made in the manuscript.

(3) page 3/lines 127-129:Modifications were made in the manuscript.

(4) page 3/lines 133-134:Modifications were made in the manuscript.

(5) page 6/lines 201-204:Modifications were made in the manuscript.

Q2. Introduction - 1st paragraph - completely missing context. The reader does not know anything about the mirror and the system.

Response:

It has been modified in the manuscript according to your comments.

Q3. Introduction - page 2/lines 57-76 - what is the purpose of this paragraph? There is a lot of references to the papers with the similar topic. The authors should clarify why they include them and how they improve the methods used in the referenced papers.

Response:

According to your comment, the paragraph has been modified accordingly.

page 2/lines 57-76: To show the readers, most scholars still use the finite element method to analyze the frequency response of the structure. This manuscript proposes the state space method to analyze the frequency response of complex structures based on the finite element analysis. In order to improve the calculation efficiency, and has a certain accuracy.

Q4. Section 2 - Principle of modal analysis - does not contain anything new. The described methods are well-known while the authors do not clarify what is new in their approach. In fact, the whole section summarizes a usual content of the advanced linear structural dynamic courses at the technical universities.

Response:

The section 2 introduces the theoretical basis of the whole manuscript. This manuscript is based on the modal analysis principle, state space model establishment theory and modal reduction theory introduced in the section 2. The frequency response analysis of the complex structure model is carried out directly through the program written in MATLAB. It is much more efficient than using the finite element method to analyze the frequency response of the structure.

Q5. Modal matrix and equations decoupling - the statements about the decoupling are misleading. It is not true that the vibration system can be, in general, decoupled as demonstrated, as the authors do not discuss the problems related to the damping matrix. Even in the example shown in Figure 2, the system can be decoupled only for certain values of damping coefficients that will produce diagonal modal damping matrix. The problem of damping is serious and damping related effects can be observed even in the presented results of the paper and are not discussed.

Response:

In the second part, the theoretical derivation part has been corrected and simplified again, and the normal mode is generally used for the analysis of the linear system with small damping. Solve the un-damped eigenvalue problem, which identifies the resonant frequencies and mode shapes (eigenvalues and eigenvectors), useful in themselves for understanding basic motions of the system. Use the eigenvectors to uncouple or diagonalize the original set of coupled equations, allowing the solution of n-uncoupled single-dof problems instead of solving a set of n-coupled equations. In general, an arbitrary damping matrix cannot be diagonalized by the un-damped eigenvectors, as the mass and stiffness matrices can. This leads to using what is called “proportional damping” in most finite element simulations. The structural damping of the primary mirror assembly adopts simple uniform damping. The second paragraph of section 3.2 describes the selection of structural damping.

Q6. Section 3.1, page 10/lines 320-323 - Boundary conditions and models are not sufficiently described. Why 10e4 point mass (shaker table experiment)? RBE2 - rigid coupling only allowing certain movements of the mirror? Why did the authors select such boundary conditions?

Response:

Large mass simulation method is used to simulate the shaker in vibration test. In an on-ground aerospace vibration test, the shaker is excited by electromagnetic coil and only vibrates in only one DOF constraint by hydrostatic slide guide. This is simulated in Patran by selecting such boundary conditions. In order to ensure the accuracy of the calculation, the large mass point is 10e3-10e8 times the structural mass. When the mass of the large mass point is set to 10e3 times the mass of the component, applying acceleration on the mass point cannot fully excite the structural response of the mirror body, and there is a rigid body mode. When the mass of the large mass point is set to 10e4 times the mass of the component, it can not only stimulate the structural response of the mirror body, but also better imitate the excitation of the vibrating table to the mirror body holder. Therefore, a quality point with a quality of 10e4 times the mass of the mirror body is established. The mass point and the reflector assembly are constrained in the form of MPC. Considering the rigid connection between the mirror body and the vibrating table, the RBE2 MPC is used to connect the large mass point unit to the node of the main mirror substrate mechanical installation interface. During the test, the connection relationship between the vibrating table and the mirror assembly remains unchanged, and there is no relative movement between the two. Therefore, the boundary condition is a constrained boundary condition, that is, the large mass point is constrained with 6 degrees of freedom and the mirror is calculated the first 10 natural frequencies.

It has been modified in the corresponding part of the manuscript.

Q7. Figure 5 - missing coordinate system.

Response:

It has been modified in the manuscript according to your comments.

Q8. Results - node numbers (FE simulation) are irrelevant for reader as he does not know the model.

Response:

It has been stated in the manuscript that node 53798 is the established mass point, that is, the incentive input point; node 60490 is the response output point; it has been marked in Figure 4 according to your comment.

Q9. Figures 7-13 - obviously, the state space model does not correspond to the FE model - what are the reasons? Numerical damping of the FE simulation? Damping of the state space model? (e.g., high frequency peak is not detectable for state space model - certainly, it is not in the location suggested by the authors).

Response:

The high frequency truncation is carried out by the state space method, and only the first ten modes are truncated for analysis. Starting from the 11th mode, the frequency is much higher than that of the 10th mode, and the response of the structural system to high-order frequencies is extremely low, so there is no need for further analysis. The frequency response analysis of the finite element method is carried out on the basis of structural modal analysis and contains high frequency information, so the curves do not coincide completely.

Q10. Experimental results - the frequency spectra are very different in comparison to the models. Authors do not discuss anything about it. Quality of the plots is very low. What is the meaning of AlarmLow curve, etc.?

Response:

In accordance with your comment, I have revised the corresponding position in the manuscript.

There are errors between the theoretical model and the actual structure. But the frequency response curve tends to be the same, and the peak response is the same. The relative error between the natural frequencies in the X/Y/Z directions obtained from the experiment and those obtained from the modal analysis is 6.7%/3.7%/6.3% respectively, which is enough to verify the accuracy of the simulation analysis.

The test pictures were scanned and then pasted into the manuscript, so the quality of the pictures was poor. I feel very sorry for this.

The Alarmlow/Alarmhigh/Abortlow/Abortion/Control curves in Figure 17 all aim at the safety control of the shaker. These curves are threshold triggers for the measured acceleration by the sensors that bonded to the shaker table. Once these triggers are touched, Alarms or Emergency Stop actions will be taken by the shaker control software to protect the shaker table mechanism or structure from resonance, collision or any other malfunctions. This is just what the control software has been set by the shaker vendor.

Q11. Table 1 - what is compared? Which simulation (FE, state space)? How the authors selected the experimental results of Eigen-frequencies? How did they determine the frequency from three coupled peaks in the experimental data?

Response:

The vibration test is to verify the accuracy of the simulation analysis. Figure 17 shows the frequency that excites the translational state in the three directions of x/y/z, that is, the frequency in the translational state along the x/y/z respectively. Compared with the corresponding modal frequencies of the main mirror assembly modal analysis mirror assembly in the x/y/z three-direction translation state, the relative error is 6.7%/3.7%/6.3% respectively, which proves that the simulation analysis results of the primary mirror assembly are accurate.

Reviewer 3 Report

The paper deals with the dynamic characteristics of a space mirror with a diameter of ϕ740mm. The authors provide a method based on the theory of modal analysis and state space theory.
The simulation tests show the effectivness of the method.
The paper is reasonably written and very interesting and useful in the context of  dynamic analysis of optical and mchanical structures. 
My suggestion is to accept the paper for publication, provided the authors arewilling to address the following suggestions:

Minor comment

  1. In (4) you use m and k. Are they scalar element? How are related to M and K matrices? Is m=M_ii? I think it’s better to clarify.
  2. In (7) and (8) c3 and k3 should be c2 and k3 respectively
  3. In (14) and (15) you use the bold style to indicate the modal and       stiffness matrices. In the paper, please keep the same symbol for objects
  4. There  are several typos and grammar errors that must be corrected (e.g. see row 158)

Author Response

Response Letter

Professor Maria Simion

Section Managing Editor

Applied Science

May 14, 2021

Dear Prof. Simion:

Subject: Submission of revised manuscript Dynamic Analysis and Experiment of Space Mirror Based on Linear State Space Expression. (Manuscript ID applcsi-1217081)

Sincerely thank you and the reviewers for the time that you have invested in evaluating our manuscript and the valuable comments. We sincerely apologize for the great time it has taken us to respond to these, and hope that a revised version of the manuscript will still be considered by Applied Science. We have modified the manuscript in response to the extensive and insightful reviewer’s comments. Our responses are given in a point-by-point manner below. Every change that we made to the manuscript are shown in another word file that we uploaded by using yellow-color-highlighted text.

Reviewer #3:

Comments to the Author:

Q1. In (4) you use m and k. Are they scalar element? How are related to M and K matrices? Is m=M_ii? I think it’s better to clarify.

Response:

It has been modified in the manuscript according to your comments.

Q2. In (7) and (8) c3 and k3 should be c2 and k3 respectively.

Response:

It has been modified in the manuscript according to your comments.

Q3. In (14) and (15) you use the bold style to indicate the modal and stiffness matrices. In the paper, please keep the same symbol for objects

Response:

It has been modified in the manuscript according to your comments.

Q4. There are several typos and grammar errors that must be corrected (e.g. see row 158).

Response:

It has been modified in the manuscript according to your comments

Round 2

Reviewer 2 Report

### REVIEW REPORT ###

MANUSCRIPT ID: applsci-1217081

MANUSCRIPT TITLE: Dynamic Analysis and Experiment of Space Mirror Based on Linear State Space Expression

RECOMMENDATION, ROUND 2: Major revision

REPORT:
To be honest, I expected that the authors will put more effort to clarify all the queries and will perform all the revisions with care. Instead, the revised version was prepared, in my opinion, by applying the minimum effort scheme. On the other hand, the revised text put the study into a wider context and defined the main aim of the paper. Although linear state space model is a known approach to obtain frequency dependent response of multi-DOF system subjected to vibrations, the papers dealing with this topic on real constructions are not very common. Also, I agree with the authors that still, the linear state space representation is not used on regularly basis and the application on space mirror is interesting. However, the article has serious flaws that have to be addressed before its publication. Therefore, I do not recommend the paper for publication in its current form and claim that the major revision is necessary.

COMMENTS:
1) Title - the original title: "Dynamic Analysis and Experiment of Space Mirror Based on Linear State Space Expression" has been changed to "Dynamic Analysis of and Experiment with a Space Mirror Based on Linear State Space Expression" which is obviously a mistake. The modified title is not highlighted in yellow color. Therefore, I strictly recommend to go through the paper again and check the whole text as well as all the images for similar problems. Also, I would strictly recommend to perform comprehensive language correction as the modified sentences are often full of repetitive words or are not grammatically correct. To make things even worse, there are some Chinese expressions throughout the article (e.g., page 12, lines 349-350) showing that the pre-submission check has not been performed with care.
2) Cover letter R1 - What changed from the previous version? I do not see any highlighted text in the corresponding places and I am not able to identify the changes elsewhere.
3) Cover letter R3 - What does the "certain accuracy" mean? The authors should clarify why to use linear state space model instead of the comprehensive FEM analysis.
4) Cover letter R4 - Linear state space model efficiency as a motivation - OK. However in that case, the authors should definitely reformulate the main message of the article and show not only comparison of the final results of eigenfrequencies. More importantly, the authors should analyze performance and calculation costs of the linear state space model and compare them with the FEA. The benefits over the FEA should be clearly presented. This is done nowhere in the article. In this context, the main results presented in Table 1 do not convince the reader about the benefits of linear state space model as the results of FEA are compared with the experiment. Better explanation and role of the FEA modal analysis for construction of the linear state space model is needed also (page 11, line 346).
5) Cover letter R5 - I understand the decoupling principle of the original set of equations to a number of single-DOF ODEs as well as the effects of damping. However, I did not identify any information about damping in the second paragraph of 3.2 as the authors stated. There are information only about the mass points and interaction of the mass-equivalent of the shaker with the mirror structure. The only information about damping are the non-highlighted statements about modal damping of 0.02 g. The authors should clarify why this value was used in the analysis.
6) Cover letter R6 - OK.
7) Cover letter R7 - The quality of the coordinate system triad in Figure 5 is rather low and hardly readable. Please, modify all the images accordingly and check image quality and readability all over the article.
8) Cover letter R9 - This is not the answer for the original question. I understand that there is a truncation of the model and that the authors limited the number of modes used in the simulation and that the high frequency response can be neglected. However, there are magnitude differences in the low frequency part of the spectrum. Comments explaining the causes of the differences are crucially needed in the article.
9) Cover letter R10 - This is not the answer for the original question. I understand that the curves from the experiment do not perfectly correspond with the model, but the authors have to explain why they are so different in shape (e.g., three overlapping peaks). More importantly, in my original note, I asked how were the peaks representing the eigenfrequencies selected. The plots have logarithmic scale over the y-axis. In some cases, there are peaks that are several magnitudes higher than those selected by the authors as the peaks representing the eigenfrequencies. How the authors determined that the lower amplitude peaks are correct? Without relevant explanation, this can be interpreted such that the peak values were selected purposively - to match with the values predicted by the model. Figure 17 should be modified to meet requirements on image clarity and quality. I understand that the plots were digitized using a scanner, but I really think that it is necessary to convert them into digital form and present them as high quality plots. As explained by the authors, the alarm thresholds can be completely omitted in the graphs as they are absolutely irrelevant for the paper.
10) Cover letter R11 - As I have already stated in point 4), the authors should compare not only the experiment with FEA but also performance and calculation costs of linear state space model to show its benefits. What was the calculation time of FEA and linear state space model? What were the computational requirements (RAM allocation, CPU usage, etc.)?

Round 3

Reviewer 2 Report

The authors answered all the questions and modified the manuscript accordingly.

Although I still think that the presentation of the experimental results and quality of some images could be improved, I don't have any factual objections. If the image quality and the form of presentation is acceptable for the journal editor, it is possible to publish the paper in its current form.